# Bandemia as an Early Predictive Marker of Bacteremia: A Retrospective Cohort Study

**DOI:** 10.3390/ijerph19042275

**Published:** 2022-02-17

**Authors:** Taku Harada, Yukinori Harada, Kohei Morinaga, Takanobu Hirosawa, Taro Shimizu

**Affiliations:** 1Department of General Medicine, Showa Koto Toyosu University Hospital, Tokyo 135-8577, Japan; hrdtaku@gmail.com; 2Department of Diagnostic and Generalist Medicine, Dokkyo Medical University Hospital, Tochigi 321-0293, Japan; yuki.gym23@gmail.com (Y.H.); k.morinaga0815@gmail.com (K.M.); t.hirosawa1983@gmail.com (T.H.)

**Keywords:** bandemia, bacteremia, clinical decision support system, blood culture results, bandemia-based electronic alert

## Abstract

This single-center retrospective observational study aimed to verify whether a diagnosis of bandemia could be a predictive marker for bacteremia. We assessed 970 consecutive patients (median age 73 years; male 64.8%) who underwent two or more sets of blood cultures between April 2015 and March 2016 in both inpatient and outpatient settings. We assessed the value of bandemia (band count > 10%) and the percentage band count for predicting bacteremia using logistic regression models. Bandemia was detected in 151 cases (15.6%) and bacteremia was detected in 188 cases (19.4%). The incidence of bacteremia was significantly higher in cases with bandemia (52.3% vs. 13.3%; odds ratio (OR) = 7.15; 95% confidence interval (CI) 4.91–10.5). The sensitivity and specificity of bandemia for predicting bacteremia were 0.42 and 0.91, respectively. The bandemia was retained as an independent predictive factor for the multivariable logistic regression model (OR, 6.13; 95% CI, 4.02–9.40). Bandemia is useful for establishing the risk of bacteremia, regardless of the care setting (inpatient or outpatient), with a demonstrable relationship between increased risk and bacteremia. A bandemia-based electronic alert for blood-culture collection may contribute to the improved diagnosis of bacteremia.

## 1. Introduction

Bacteremia is a form of severe infection associated with a high probability of mortality and substantial financial healthcare costs [1].

Bacteremia is a poor prognostic factor for infectious diseases and is one of the leading causes of death in developed countries [2,3]. Specifically, the 1-month rate of community-acquired bacteremia is 10–19%, and the mortality rate for hospital-acquired bloodstream infection is 17–28% [4,5,6,7]. Therefore, the early recognition of bacteremia is clinically important because it could assist in the faster initiation of proper treatment, leading to improved clinical outcomes. In previous studies, the experience of a “chill” has been focused on as a simple and effective predictive factor for bacteremia [8,9,10,11,12,13,14,15], and several prediction rules, which were created by merging multivariate logistic regression models, have been suggested [16,17,18]. The only single highly predictive factor of bacteremia was a shaking chill, and even then, the prevalence of bacteremia in these cases was only 27% [1,12]. The clinical rule described by Shapiro et al. is highly sensitive in nature; its external validity has been verified and is useful for preventing the misdiagnosis of bacteremia [19,20]. However, its specificity is 29–48%, whilst the prevalence of bacteremia is only 15–37%; besides, the clinical rule requires 11 items of clinical information. Experiencing chills and most of the symptoms described in the existing prediction models are based on patients’ subjective complaints, all of which require a mastery of interpretation of symptoms, which are highly subjective in nature. Additionally, physicians inaccurately predict—and often overestimate—the risk of bacteremia [21]. Therefore, based on certain factors, there is a need for models that can predict the presence of bacteremia, without being influenced by the subjectivity of symptoms that are difficult to accurately assess or may be affected by physician bias.

The clinical decision support system (CDSS) has been demonstrated to improve the quality of medical care in various medical settings; however, its usefulness in diagnosing acute illnesses has not yet been verified [22]. Most of the studies on CDSS in bacteremia were aimed at management and surveillance, and the only report on the prediction of bacteremia by the CDSS is “TREAT” by Paul et al. [23]. TREAT is unique in that it stratified the risk of bacteremia in patients and incorporated 11 different infection sites and 34 different diagnoses, with a 28% prevalence of bacteremia in the highest risk group. Some studies on machine learning for predicting bacteremia have been conducted; however, the accuracy is still limited [24].

Bandemia is a condition in which 10% or more counts of band cells (a type of immature neutrophil) are observed in peripheral blood smears, being a potentially useful objective predicting factor for bacteremia [25]. Indeed, the results of studies investigating bandemia as a useful marker for predicting bacteremia, reported in several different settings, have been promising [16,26,27,28,29].

Previous CDSS studies on bandemia are unavailable. We hypothesized that a system to generate blood-culture alerts, triggered by new onset bandemia, could contribute to the early diagnosis of bacteremia if incorporated into the electronic medical records; additionally, we tested the association between the presence of bandemia and bacteremia using retrospective data.

## 2. Materials and Methods

### 2.1. Patient Selection

This was a single-center retrospective observational study. We included consecutive outpatients and inpatients who underwent two or more sets of blood cultures and a complete blood count, with a differential white blood cell (WBC) test. Tests were taken 24 h from the time of the patients’ presentation in an outpatient facility and the onset of fever in the inpatient facility at Dokkyo Medical University Hospital between 1 April 2015, and 31 March 2016. The exclusion criteria were developed to exclude factors that can affect WBC differentials or blood culture results and were as follows: age ≤ 14 years, antibiotic usage the day prior to blood sampling, use of granulocytecolony stimulating factors within 1 week, use of steroids and immunosuppressants within 1 week, chemotherapy for cancer within 4 weeks, history of hematological malignancy or bone metastasis of a solid tumor, irradiation therapy for a malignant tumor, pregnancy, systemic lupus erythematosus, post-resuscitation, human immunodeficiency virus infection, or contamination cases. This study was approved by the Ethics Committee of Dokkyo Medical University Hospital and conducted in accordance with the Declaration of Helsinki. The requirement for written informed consent was waived due to the retrospective study design. 

### 2.2. Data Collections, Outcomes, and Definitions

Blood culture and WBC differential tests were performed at the discretion of the examining doctors and subsequently performed by an automated analyzer. The sample of the blood cell count were collected by drawing blood into a tube containing an ethylenediaminetetraacetic acid dipotassium. The measurements of the blood cell count, including band count, were analyzed using the Sysmex XN-2000 automatic blood cell analyzer.

We extracted data regarding age, sex, body temperature, WBC count, eosinophil count, percentage band count in the differential WBC test, and blood culture results from the electronic medical records. The primary outcome measurement was bacteremia, which was defined as a positive blood culture 1 week after the samples were collected [20]. We defined contamination as the presence of multiplying coagulase-negative Staphylococcus species, Bacillus species, Propionibacterium acnes, or Corynebacterium species in a single set of blood cultures [30]. Bandemia was defined as a band count of >10%, based on a previous study [8]. Eosinopenia, which has been reported as a risk of bacteremia, was set at a cutoff of 25 cells/mm^3^ [31].

### 2.3. Statistical Analysis

Continuous variables are presented as medians, with 25th and 75th percentiles, and categorical variables as counts (percentage). Continuous variables were compared using the Wilcoxon rank-sum test. Categorical variables were compared by using chi-square test. Univariate and multivariate logistic regression models were used to assess the predictive values of bandemia. We created two multivariate logistic regression models as follows: the first included age (over 65 or not), sex, body temperature (between 36.0 and 38.0 or not), WBC count (between 4000 and 12,000 or not), and eosinopenia as variables (baseline model); the second added bandemia as a variable to the baseline model (with band model). Furthermore, we calculated the area under the receiver-operating characteristic curve (AUROC) of each model with a 95% confidence interval (CI). We compared the AUROCs of the above-mentioned two models, calculated the net reclassification improvement (NRI) and integrated discrimination improvement (IDI) to assess the improvement of the predictive value after including the percentage band count into a baseline model. The *p*-values for statistical tests were two-tailed, and a *p*-value < 0.05 was considered statistically significant. All statistical analyses were performed using R version 3.6.1 (The R Foundation for Statistical Computing, Vienna, Austria).

## 3. Results

In total, 4879 cases during the study period were considered eligible and 970 cases (inpatient, 559 cases; outpatient, 411 cases) were included in the final analysis. The median age was 73 (range, 63–80) years; 629 (64.8%) were men; median body temperature was 38.1 (range, 37.2–38.6) °C; median white blood cell (WBC) count was 10,600 (range, 7800–11,790) cells/mm^3^; median eosinophil count was 20 (range, 0–93) cells/mm^3^; and the median percentage band count was 0% (range, 0–2%) (Table 1). Overall, bandemia was detected in 151 cases (15.6%) and bacteremia in 188 cases (19.4%). Among the 188 cases with bacteremia, 190 micro-organisms were isolated (Table 2). The incidence of bandemia was higher in inpatients than in outpatients (inpatients, 18.4%; outpatients, 11.7%; *p* = 0.004), while there was no significant differences of the incidence of bacteremia between inpatients and outpatients (inpatients, 19.0%; outpatients, 20.0%; *p* = 0.700).

The unadjusted incidence of bacteremia was significantly higher among patients with bandemia than in those without bandemia (52.3% vs. 13.3%; OR, 7.15; 95% CI, 4.91–10.5; *p* < 0.001). The sensitivity and specificity of bandemia for predicting bacteremia were 0.42 (95% CI, 0.35–0.49) and 0.91 (95% CI, 0.89–0.93), respectively. This was consistent in outpatients (52.1% vs. 15.7%; OR, 5.80; 95%CI, 2.94–11.53; *p* < 0.001) and inpatients (52.4% vs. 11.4%; OR, 8.51; 95% CI, 5.12–14.26; *p* < 0.001). The bandemia was retained as an independent predictive factor in a multivariable logistic regression model (OR, 6.13; 95% CI, 4.02 to 9.40; *p* < 0.001; Table 3).

Furthermore, the AUROC of the prediction model for bacteremia was statistically improved from 0.71 (95% CI 0.67–0.75) to 0.77 (95% CI 0.73–0.80) in the band-count model compared to the baseline model (Figure 1). Similarly, the NRI (0.62; 95% CI 0.47–0.76; *p* < 0.01) and IDI (0.09; 95% CI 0.07–0.12; *p* < 0.01) were statistically significant.

## 4. Discussion

This study presents three main findings. First, in patients considered as having indicators for blood cultures by a physician, the prevalence of bacteremia was significantly higher among patient with than without bandemia. Second, bandemia revealed a high specificity and OR for predicting bacteremia. Third, the band-count percentage as a continuous variable, was an independent predictor for bacteremia, and the AUROC was improved by adding the band-count percentage as a continuous variable to the baseline predictive model including age, sex, body temperature, WBC count, and eosinophil count. Our findings support the value of bandemia for predicting bacteremia among patients with a suspected bacterial or fungal infection. 

Consistent data from recent studies indicate bandemia (band count > 5% or 10%) as an independent predictor of bacteremia in patients who presented to the emergency department [17,28] and were admitted to general wards [26] or to an intensive care unit [29]. Our study confirms the value of bandemia for the prediction of bacteremia in a more extended population of patients who presented to outpatient department or inpatients who developed a fever after admission. Additionally, our study suggests that the predictive value of bandemia seems to be independent of the total white blood cell count. Even after adjustment and including other factors, such as the WBC count, the OR of band count percentage for predicting bacteremia was stable; previous study results corroborate this finding [16,26]. In a retrospective study that included only patients with a normal WBC count on admission, bandemia (band count > 10%) was a strong predictor of bacteremia [16]. Another prospective cohort study also showed that bandemia (band count > 5%) was observed in 79% of patients with bacteremia who had a normal WBC count [28]. In addition, the band-count percentage was not associated with the total WBC count in patients with sepsis [29]. Moreover, the study reported that the band count percentage, rather than the WBC count, could distinguish patients with sepsis from those with non-infectious systemic inflammatory response syndrome [29]. Therefore, bandemia could be a useful marker for predicting bacteremia irrespective of the WBC count.

The strengths of this study include its demonstration of bandemia as a risk factor for bacteremia in both inpatient and outpatient settings and the use of machine-determined bandemia techniques, leading to an easy and objective determination of results. The advantage of bandemia is that the presence or absence of findings can be determined inexpensively, using labor-saving and labor-free machine counts. In recent years, new techniques such as an antibody microarray coupled to a Surface Plasmon Resonance imager (SPRi), PCR methods and Fluorescent in situ hybridization (FISH), were developed [32,33,34,35]. These techniques are labor-intensive and costly at the moment; future developments may advance diagnostic strategies for bacteremia. Additionally, there is a proportional relationship between the extent of bandemia and the risk of bacteremia. In addition, 52.3% of the patients in the bandemia group had bacteremia, and the presence of bandemia led to the prediction of bacteremia with a sensitivity and specificity of 0.42 and 0.91, respectively. Studies on clinical prediction rules for bacteremia are characterized by high sensitivity and low specificity because they focus on the presence of bacteremia [17,19,20]. The prediction of blood cultures by machine learning using Neural Networks has a specificity and sensitivity of 88% and 17%, respectively [24]. Therefore, if E-alert (which recommends the collection of blood cultures based on a diagnosis od bandemia) is incorporated, it could be well-received by clinicians because of its high specificity and possible assistance in the timely diagnosis of bacteremia.

There are several limitations to this study which should be acknowledged in the interpretation of the results for practice. Th most important limitations are the retrospective study design, the lack of specific criteria indicating the collection of blood cultures (selection bias could, therefore, not be avoided), and the non-inclusion of other objective variables in the analysis, such as vital signs, owing to the lack of data. Second, the conditions under which CBCs were collected may not be constant. Whether the patient is fasting or not, the time of collection, and the location of the blood collection were not included in this analysis. Third, this is not a prospective study of CDSS, although we have shown that bandemia is a useful trigger for noticing bacteremia in both in- and out-patients. Fourth, this study excluded children and, therefore, may not be applicable to pediatric populations. Lastly, our data were derived from a single tertiary academic hospital. As such, the study population tended to present serious medical conditions which, arguably, could be the reason for the high incidence of bacteremia. Additionally, several patients were omitted based on the exclusion criteria. Therefore, the study results should be applied with caution in patients who present with at least one of the exclusion criteria. The next step is to validate our results in a prospective CDSS study that incorporates the exclusion criteria.

## 5. Conclusions

In conclusion, this study demonstrated that bandemia is a useful predictor for bac-teremia, regardless of the care setting, and that there is a correlation between bandemia and an increased risk of bacteremia in adults. Therefore, the inclusion of bandemia-based electronic alerts or CDSS for blood-culture collection may improve the diagnosis of bacteremia in both outpatient and inpatient settings in adults.

## Figures and Tables

**Figure 1 ijerph-19-02275-f001:**
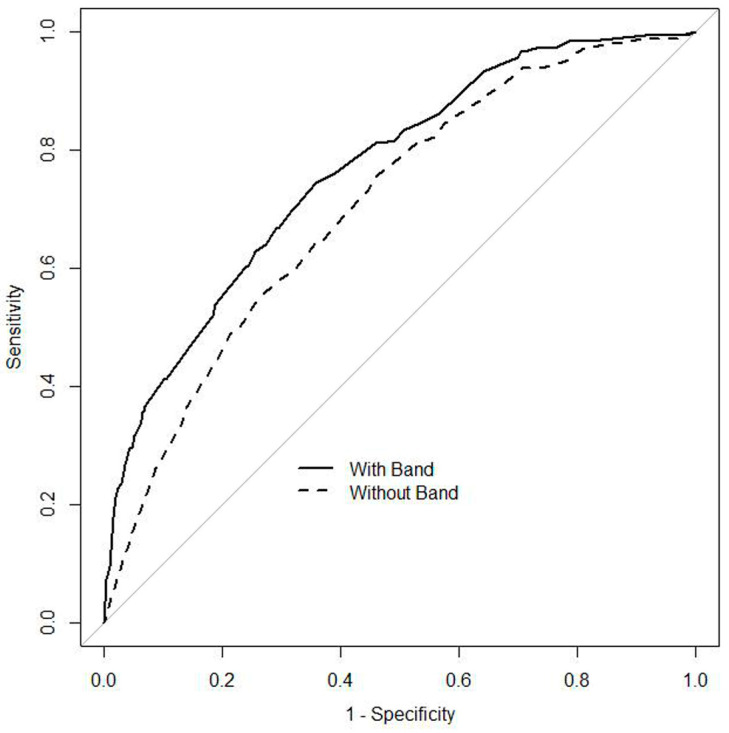
Receiver operating characteristic curves of multivariable logistic regression models for predicting bacteremia. The ‘With Band’ curve includes age, sex, body temperature, WBC count, eosinophil count, and band count percentage, while the ‘Without Band’ curve includes age, sex, body temperature, WBC count, and eosinophil count.

**Table 1 ijerph-19-02275-t001:** Characteristics of the distinguished groups (bacteremia and no bacteremia).

	Bacteremia (N = 188)	No Bacteremia (N = 782)	*p*-Value
Age > 65 (%)	143/188 (76.1%)	530/782 (67.8%)	0.027
Men (%)	110/188 (58.5%)	519/782 (66.4%)	0.043
Body temperature * >38.0 or <36.0 (%)	135/180 (75.0%)	368/735 (50.1%)	0.001
White blood cell count >12,000 or <4000 cells/mm^3^ (%)	100/188 (53.2%)	313/782 (40.0%)	<0.001
Eosinophil count <25 cells/mm^3^ (%)	135/188 (71.8%)	379/782 (48.5%)	<0.001
Bandemia (%)	79/188 (42.0%)	72/782(9.2%)	<0.001

* n = 915.

**Table 2 ijerph-19-02275-t002:** Micro-organisms isolated in bacteremia.

Microbiology	Isolate Number (N = 190)
*Escherichia coli*	52 (27.4%)
*Staphylococcus aureus*	37 (19.5%)
*Klebsiella pneumonia*	26 (13.7%)
*Klebsiella oxytoca*	9 (4.7%)
*Enterococcus faecalis*	8 (4.2%)
Coagulase-negative staphylococcus	7 (3.7%)
*Enterobacter cloacae*	6 (3.2%)
*Pseudomonas aeruginosa*	6 (3.2%)
*Streptcoccus pneumonia*	6 (3.2%)
*Streptcoccus agalactiae*	4 (2.1%)
*Bacteroides* spp.	4 (2.1%)
*Serratia marcescens*	3 (1.6%)
Group G streptococci	3 (1.6%)
*Other Streptococcus* spp.	6 (3.2%)
Miscellaneous	13 (6.8%)

**Table 3 ijerph-19-02275-t003:** Odds ratios of variables for predicting bacteremia.

	Odds Ratio (Univariate Model)	*p*-Value	Odds Ratio (Multivariate Model)	*p*-Value
Bandemia	7.15 (4.91–10.50)	<0.001	6.13 (4.02–9.40)	<0.001
Age > 65	1.51 (1.05–2.20)	0.028	1.46 (0.97–2.22)	0.075
Male	0.71 (0.52–0.99)	0.043	0.86 (0.59–1.24)	0.411
Body temperature >38.0 or <36.0	2.99 (2.09–4.36)	<0.001	3.22 (2.18–4.84)	<0.01
White blood cell count >12,000 or <4000 cells/mm^3^	1.70 (1.24–2.35)	<0.001	1.15 (0.79–1.66)	0.471
Eosinophil count <25 cells/mm^3^	2.71 (1.92–3.86)	<0.001	1.99 (1.35–2.97)	0.001

Data are shown as odds ratio (95% confidence interval). Odds ratios and *p*-values are derived from logistic regression models.

## Data Availability

The datasets generated and/or analyzed during the current study are not publicly available as the ethics committee did not provide permission, but they are available from the corresponding author upon reasonable request.

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
