# Peer review of "Bandemia as an Early Predictive Marker of Bacteremia: A Retrospective Cohort Study"

_ijerph, 2022, doi:10.3390/ijerph19042275_

Round 1

Reviewer 1 Report

In their work, the authors addressed the important topic of the prediction of bacteremia. Bandemia (>10%) is a  parameter available in routine laboratory diagnostics, closely related to the development of infection.  Importantly, the predictive utility of this marker is indicated even with the normal leukocyte counts.

The paper is written in a very legible and concise manner, but requires minor corrections and additional information, as listed below:

Material and methods:

  1. It is necessary to indicate the hematological analyzer used for the CBC, in particular the manufacturer of the equipment and technology in which the test is performed. Currently, various technologies based on different reagents are used on the market, therefore the obtained results may differ from each other. All patient examinations were performed with the same technology?
  2. Please describe in more detail the conditions of blood donation from patients. Was the CBC always performed under the same conditions, e.g. under fasting conditon,  at a specific time, always from a vein in the arm? What anticoagulant was used to  blood collection? If there was no such standardization, please take it into account in your work.

Statistical analysis:

  1. The assumptions of logistic regression require a normal distribution of quantitative data in particular groups. Has this assumption been met? If such assumptions were met, it is more appropriate to present the data as mean and standard deviation. Please provide relevant information at work.

Results:

  1. lines 118-132. The description of the contents of Table 1 is inconsistent with the facts. Table 1 presents the characteristics of the distinguished groups ( bacteremia and no bacteremia), not the characteristics of the entire study population.
  2. lines 124-125. It was not specified what statistical test was used to compare the incidence of bacteremia in the analyzed groups and with what probability the null hypothesis was rejected.
  3. Table 1. I suggest entering the P-value uniformly to three decimal places 
  4. Figure 1. It would be valuable to repeat the information on the parameters used to create the baseline model when discussing the ROC curve in the Results section (in the text preceding figure 1 or as a legend to this figure).

Discussion:

  1. lines 175-177.  It does not appear from the content of the study that bandemia is an equally good marker for the population inpatent as well as outpatient, because the authors did not make such a comparison. At the same time, due to the retrospective nature of the work, the authors did not demonstrate its usefulness with the CDSS.

Conclusions:

1. The conclusions should indicate the correctness of the inference limited to the population of people over 14 years of age, because that was the studied population. The fact of different values of the differential WBC test in children is well known.

Author Response

Thank you for your careful review and kind consideration of our manuscript titled “Bandemia as an early predictive marker of bacteremia: a retrospective cohort study"

We appreciate the invaluable comments that the reviewers provided, which we are confident have helped us improve the manuscript. We have provided point-by-point responses to each of the reviewers’ comments and describe the related revisions below. As per your instructions, we have indicated all changes via red characters in the revised manuscript.

Thank you for your consideration.
We look forward to any further comments regarding the revised manuscript.

Sincerely yours,

Taku Harada, MD
Division of General Medicine, Showa University Koto Toyosu Hospital
5-1-38 Toyosu, Koto-ku, Tokyo 135-8577 Japan
Phone: +81-3-6204-6000, FAX: +81-3-6204-6396
E-mail: hrdtaku@gmail.com

â– For Reviewer1

In their work, the authors addressed the important topic of the prediction of bacteremia. Bandemia (>10%) is a parameter available in routine laboratory diagnostics, closely related to the development of infection.  Importantly, the predictive utility of this marker is indicated even with the normal leukocyte counts.

The paper is written in a very legible and concise manner, but requires minor corrections and additional information, as listed below:

Comment:
Material and methods:

It is necessary to indicate the hematological analyzer used for the CBC, in particular the manufacturer of the equipment and technology in which the test is performed. Currently, various technologies based on different reagents are used on the market, therefore the obtained results may differ from each other. All patient examinations were performed with the same technology?
Please describe in more detail the conditions of blood donation from patients. Was the CBC always performed under the same conditions, e.g. under fasting conditon,  at a specific time, always from a vein in the arm? What anticoagulant was used to  blood collection? If there was no such standardization, please take it into account in your work.

Response:
Thank you for your constructive comments.
In response to your comments, we have added the manuscript as follows

P2L92-P3 L95

The sample of blood cell count were collected by drawing blood into a tube containing an ethylenediaminetetraacetic acid dipotassium. Measurement of blood cell count including band count were analyzed using the Sysmex XN-2000 automatic blood cell analyzer.

P7 L241-3

Second, the conditions under which CBCs were collected may not be constant. Whether the patient is fasting or not, the time of collection, and the location of the blood collection are not included in this analysis

Comment:
Statistical analysis:
The assumptions of logistic regression require a normal distribution of quantitative data in particular groups. Has this assumption been met? If such assumptions were met, it is more appropriate to present the data as mean and standard deviation. Please provide relevant information at work.

Resnponse:

Thank you for your constructive comments. No variables in this study met normal distribution, therefore, all continuous variables were presented as median (25-75 percentiles), and compared by using Wilcoxon rank sum test. Regarding the logistic regression analysis, we performed logistic regression analyses by including variables which were converted from continuous to binary (e.g. age was converted to age > 65 or not). The cut-off values for converting continuous to binary variables were decided based on the results of previous studies and SIRS criteria. Bandemia was still an independent predictive factor for bacteremia in these new analyses, and other main results were unchanged.

We have revised the manuscript as follows:

P3 L111-5

We created two multivariate logistic regression models as follows: the first included age (over 65 or not), sex, body temperature (between 36.0 and 38.0 or not), WBC count (between 4,000 and 12,000 or not), and eosinopenia as variables (baseline model); the second added bandemia as a variable to the baseline model (with band model).

P4 L166-L5 180

The bandemia was retained as an independent predictive factor on multivariable logistic regression model (OR, 6.13; 95% CI, 4.02 to 9.40; P<0.001; Table 3).

Table 3. Odds ratios of variables for predicting bacteremia.

Odds ratio

(Univariate model)

P-value

Odds ratio

(Multivariate model)

P-value

Bandemia

7.15 (4.91–10.50)

<0.001

6.13 (4.02–9.40)

<0.001

Age >65

1.51 (1.05–2.20)

0.028

1.46 (0.97–2.22)

0.075

Male

0.71 (0.52–0.99)

0.043

0.86 (0.59–1.24)

0.411

Body temperature

>38.0 or <36.0

2.99 (2.09–4.36)

<0.001

3.22 (2.18–4.84)

<0.01

White blood cell count

>12,000 or <4,000 cells/mm3

1.70 (1.24–2.35)

<0.001

1.15 (0.79–1.66)

0.471

Eosinophil count

<25 cells/mm3

2.71 (1.92–3.86)

<0.001

1.99 (1.35–2.97)

0.001

Data are shown as odds ratio (95% confidence interval). Odds ratios and P-values are derived from logistic regression models.

Furthermore, the AUROC of the prediction model for bacteremia was statistically improved from 0.71 (95% CI 0.67–0.75) to 0.77 (95% CI 0.73–0.80) in the band count model compared to the baseline model (Figure 1). Similarly, the NRI (0.62; 95% CI 0.47–0.76; P<0.01) and IDI (0.09; 95% CI 0.07–0.12; P<0.01) were statistically significant.

Comment:
Results:
lines 118-132. The description of the contents of Table 1 is inconsistent with the facts. Table 1 presents the characteristics of the distinguished groups ( bacteremia and no bacteremia), not the characteristics of the entire study population.

Response:
Thank you for your constructive comments.
In response to your comments, we have revised the description of the contents of Table 1 as follows

From

Table 1. Baseline characteristics.

To

Table 1. Characteristics of the distinguished groups (bacteremia and no bacteremia).

Comment:
lines 124-125. It was not specified what statistical test was used to compare the incidence of bacteremia in the analyzed groups and with what probability the null hypothesis was rejected.

Response:
Thank you for your constructive comments.

In response to your comments, we have revised the manuscript as follows:

P3 L 109-110

Categorical variables were compared by using chi-square test.

Comment:

Table 1. I suggest entering the P-value uniformly to three decimal places

Response:
Thank you for your constructive comments.
In response to your comments, I have revised the table.

Comment:

Figure 1. It would be valuable to repeat the information on the parameters used to create the baseline model when discussing the ROC curve in the Results section (in the text preceding figure 1 or as a legend to this figure).

Response:
Thank you for your constructive comments.
In response to your comments, we have added the description of the contents of Figure 1 as follows

The ‘With Band’ curve includes age, sex, body temperature, WBC count, eosinophil count, and band count percentage; The ‘Without Band’ curve includes age, sex, body temperature, WBC count, and eosinophil count.

Comment:
Discussion:
lines 175-177.  It does not appear from the content of the study that bandemia is an equally good marker for the population inpatent as well as outpatient, because the authors did not make such a comparison. At the same time, due to the retrospective nature of the work, the authors did not demonstrate its usefulness with the CDSS.

Response:
Thank you for your constructive comments.
In response to your comments, we have added the manuscript as follows:

P3 L131-4

The incidence of bandemia was higher in inpatients than in outpatients (inpatients, 18.4%; outpatients, 11.7%; P=0.004), while there was no significant differences of the incidence of bacteremia between inpatients and outpatients (inpatients, 19.0%; outpatients, 20.0%; P=0.700).

P7 L243-5

Third, this is not a prospective study of CDSS, although we have shown that shingles is a useful trigger for noticing bacteremia in both in- and out-patients.

Comment:

Conclusions:
1. The conclusions should indicate the correctness of the inference limited to the population of people over 14 years of age, because that was the studied population. The fact of different values of the differential WBC test in children is well known.

Thank you for your constructive comments.
In response to your comments, we have added the manuscript as follows (in discussion) :

P6 L245-6

Fourth, this study excluded children and, therefore, may not be applicable to pediatric populations.

Thank you for your constructive comments.
In response to your comments, I have revised the manuscript as follows (I conclusion) :

P7 L255-9

From

In conclusion, this study demonstrated that bandemia is a useful predictor for bac-teremia regardless of the care setting, and that there is a correlation between bandemia and an increased risk of bacteremia. Therefore, the inclusion of bandemia-based electronic alerts or CDSS for blood culture collection may improve the diagnosis of bacteremia in both outpatient and inpatient settings.

To

In conclusion, this study demonstrated that bandemia is a useful predictor for bac-teremia regardless of the care setting, and that there is a correlation between bandemia and an increased risk of bacteremia in adults. Therefore, the inclusion of bandemia-based electronic alerts or CDSS for blood culture collection may improve the diagnosis of bacteremia in both outpatient and inpatient settings in adults..

Reviewer 2 Report

the title is clear, the methodology is well established. The context about the bandemia associated with the bacteremia is not interested to the reader because I think we already had strong evidence about that. However, the overall manuscript is well organized. 

Author Response

Thank you for your careful review and kind consideration of our manuscript titled “Bandemia as an early predictive marker of bacteremia: a retrospective cohort study"

We appreciate the invaluable comments that the reviewers provided, which we are confident have helped us improve the manuscript. We have provided point-by-point responses to each of the reviewers’ comments and describe the related revisions below. As per your instructions, we have indicated all changes via red characters in the revised manuscript.

Thank you for your consideration.
We look forward to any further comments regarding the revised manuscript.

Reviewer 3 Report

Dear authors, the data presented in the manuscript entitled "Bandemia as an early predictive marker of bacteremia: a retrospective cohort study" have a scientific soundness among physicians. However, i have a big concern regarding that nowdays there are many instruments that can make an early prediction of bacteremia e.g BACTEC, antibody microarray coupled to a Surface Plasmon Resonance imager (SPRi), PCR methods, Fluorescent in situ hybridization (FISH); so why the authors used Bandemia which isn't specific as mentioned by the authors "The study limitations include the retrospective study design, the lack of specific criteria indicating the collection of blood cultures (selection bias could, therefore, not be avoided), and the non-inclusion of other objective variables in this analysis, such as vital signs, owing to the lack of data". 

Author Response

Thank you for your careful review and kind consideration of our manuscript titled “Bandemia as an early predictive marker of bacteremia: a retrospective cohort study"

We appreciate the invaluable comments that the reviewers provided, which we are confident have helped us improve the manuscript. We have provided point-by-point responses to each of the reviewers’ comments and describe the related revisions below. As per your instructions, we have indicated all changes via red characters in the revised manuscript.

Thank you for your consideration.
We look forward to any further comments regarding the revised manuscript.

Sincerely yours,

Taku Harada, MD
Division of General Medicine, Showa University Koto Toyosu Hospital
5-1-38 Toyosu, Koto-ku, Tokyo 135-8577 Japan
Phone: +81-3-6204-6000, FAX: +81-3-6204-6396
E-mail: hrdtaku@gmail.com

â– Reviewer3

Dear authors, the data presented in the manuscript entitled "Bandemia as an early predictive marker of bacteremia: a retrospective cohort study" have a scientific soundness among physicians. However, i have a big concern regarding that nowdays there are many instruments that can make an early prediction of bacteremia e.g BACTEC, antibody microarray coupled to a Surface Plasmon Resonance imager (SPRi), PCR methods, Fluorescent in situ hybridization (FISH); so why the authors used Bandemia which isn't specific as mentioned by the authors "The study limitations include the retrospective study design, the lack of specific criteria indicating the collection of blood cultures (selection bias could, therefore, not be avoided), and the non-inclusion of other objective variables in this analysis, such as vital signs, owing to the lack of data".

Thank you for your constructive comments.
In response to your comments, I have added the manuscript as follows

P6 L221-6

The advantage of bandemia is that the presence or absence of findings can be determined with a labor-saving, inexpensive, and labor-free machine count. In recent years, new techniques such as antibody microarray coupled to a Surface Plasmon Resonance imager (SPRi), PCR methods and Fluorescent in situ hybridization (FISH) have been developed [32-35]. These techniques are labor-intensive and costly at the moment, but future developments may advance diagnostic strategies for bacteremia.

Reviewer 4 Report

In this study, the authors investigated bandemia as a predictor for bacteremia in inpatients and outpatients. The study is well designed; the results support the conclusion. However, I have several concerns to be addressed as follows:

  • Methods: Please define the type and manufacturer name of the automated haematology analyzer.
  • The band cell count expressed as a percentage does not accurately represent the band cell count. Please consider using the absolute count of band cells instead.
  • Statistical analysis: the continuous variables are presented as medians with 25th and 75th percentiles. However, it is not clear whether you tested them first for normality. I noticed that some variables, like body temperature, seem to be normally distributed, so this should be presented as a mean and standard deviation.
  • Results: The flow of patients (screened, excluded, and finally included) is better demonstrated in a flow chart.
  • Results: other laboratory markers for bacteremia should be collected and involved in the regression model as CRP, LDH, procalcitonin, etc.
  • Results: The results of blood culture should be presented with respect to the causative organisms.
  • The headings of tables should be revised to be more informative,
  • Figure 1: Please accurately identify the models as baseline model and with band model.
  • The manuscript should be edited for the correction of some structural, grammatical, and punctuation errors.  

Author Response

Thank you for your careful review and kind consideration of our manuscript titled “Bandemia as an early predictive marker of bacteremia: a retrospective cohort study"

We appreciate the invaluable comments that the reviewers provided, which we are confident have helped us improve the manuscript. We have provided point-by-point responses to each of the reviewers’ comments and describe the related revisions below. As per your instructions, we have indicated all changes via red characters in the revised manuscript.

Thank you for your consideration.
We look forward to any further comments regarding the revised manuscript.

Sincerely yours,

Taku Harada, MD
Division of General Medicine, Showa University Koto Toyosu Hospital
5-1-38 Toyosu, Koto-ku, Tokyo 135-8577 Japan
Phone: +81-3-6204-6000, FAX: +81-3-6204-6396
E-mail: hrdtaku@gmail.com

â– Reviewer4

In this study, the authors investigated bandemia as a predictor for bacteremia in inpatients and outpatients. The study is well designed; the results support the conclusion. However, I have several concerns to be addressed as follows:

Comment:
Methods: Please define the type and manufacturer name of the automated haematology analyzer.

Response:

Thank you for your constructive comments.
In response to your comments, I have added the manuscript as follows

P2 L92-95

The sample of blood cellcount were collected by drawing blood into a tube containing an ethylenediaminetetraacetic acid dipotassium.Measurement of blood cell count including band count were analyzed using the Sysmex XN-2000 automatic blood cell analyzer.

Comment:
The band cell count expressed as a percentage does not accurately represent the band cell count. Please consider using the absolute count of band cells instead.

Response:

Thank you for your constructive comments. Eosinophils, another type of white blood cell, are classified according to the severity of the disease based on their count. However, bandemia has historically been described based on percentages in the previous literature, so we decided to use percentages in this study, including the definition.

Comment:
Statistical analysis: the continuous variables are presented as medians with 25th and 75th percentiles. However, it is not clear whether you tested them first for normality. I noticed that some variables, like body temperature, seem to be normally distributed, so this should be presented as a mean and standard deviation.

Response:

Thank you for your constructive comments.

In response to your comments, We tested again for distribution, but normality was rejected. Therefore, the median and quartiles have been included.

Comment:
Results: The flow of patients (screened, excluded, and finally included) is better demonstrated in a flow chart.

Response:

Thank you for your advice

We tried to create the flowchart you requested, but we realized that the data itself before the exclusion of this paper was lost due to the file corruption, so we found it difficult to create the flowchart.

Comment:
Results: other laboratory markers for bacteremia should be collected and involved in the regression model as CRP, LDH, procalcitonin, etc.

Response:

Thank you for your constructive comments.

Adding CRP, LDH, and PCT to the model would be worth considering. However, these variables have a risk of collinearity with bandemia due to their medical nature. Moreover, in many cases, CRP and PCT were not measured in the present data, making it difficult to adapt from the missing data perspective.

Comment:

Results: The results of blood culture should be presented with respect to the causative organisms.

Response:

Thank you for your constructive comments.
In response to your comments, I have added the manuscript and table 3 as follows:

P3 L130-1

In 188 cases of bacteremia, 190 microorganisms were isolated. The main results are shown in Table 3.

P4 L139
Table2. Microorganisms isolated in bacteremia

Comment:

Figure 1: Please accurately identify the models as baseline model and with band model.

Response:
Thank you for your constructive comments.
In response to your comments, we have added the description of the contents of Figure 1 as follows

The ‘With Band’ curve includes age, sex, body temperature, WBC count, eosinophil count, and band count percentage; Without Band: comprising age, sex, body temperature, WBC count, and eosinophil count

Comment:

The headings of tables should be revised to be more informative,

Comment:

The manuscript should be edited for the correction of some structural, grammatical, and punctuation errors.  

Response:

Thank you for your constructive comments.
In response to your comments, We asked the proofreader to revise the manuscript again, and also checked with the co-author.

Round 2

Reviewer 4 Report

The authors have adequately addressed all my concerns and queries and I have no additional comments.